# Improving Plasticity of Ferritic Stainless Steel Welded Joints via Laser Spot Control

**DOI:** 10.3390/mi14112072

**Published:** 2023-11-08

**Authors:** Lidong Gu, Qi Tang, Yanqing Li, Fengde Liu, Piyao Liu

**Affiliations:** School of Mechanical and Electrical Engineering, Changchun University of Science and Technology, Changchun 130022, China; gulidong@cust.edu.cn (L.G.); m15058623838@163.com (Q.T.); 18844116299@163.com (F.L.); 2020100619@mails.cust.edu.cn (P.L.)

**Keywords:** ferritic stainless steel, welded joint, microstructure, plasticity, laser spot control

## Abstract

The plasticity of welded 441 ferritic stainless steel joints was controlled by varying the laser beam spot diameter during laser welding. A stainless steel plate thickness of 1.2 mm was used. The microstructures of the welded joints were analyzed for various spot diameters. The elongation of breaks and the reduced area of tensile specimens were analyzed to study the effects of spot diameter on plasticity of the welded joints. The results showed that the weld melt width and weld column crystal size increased with the spot diameter, but isometric crystals in the center of the weld were gradually reduced. Increasing spot diameter resulted in decreased tensile strength, elongation after breaks, and area reduction. So the parameter must be controlled during the laser welding.

## 1. Introduction

Ferritic stainless steel (FSS) has the advantages of a low expansion coefficient, better corrosion and oxidation resistance, and low nickel content. It is gradually replacing austenitic stainless steel and is widely used in manufacturing [1]. However, during welding of FSS, the ferrite grains grow significantly, leading to problems such as grain coarsening of welded joints, cracks, and embrittlement [2], which reduce the mechanical properties of joints and limit applications. Laser welding uses high-energy-density laser beams as a heat source, which has the advantages of fast welding speeds, low heat input, and high energy density. It can inhibit grain growth and improve mechanical properties of welded joints. Zhao [3] studied the effect of ultrasonic impact treatment (UIT) on the organization and performance of laser-welded 1.5 mm 441 FSS sheets. The results show that after UIT, an ultrasonically induced reinforcing layer is produced on the butt surface of the joint, the tensile strength of the specimen is improved, and the residual stress of the welding specimen is reduced, which improves the mechanical properties. Liu [4] investigated the effects of average power and pulse duration of pulsed laser welding on weld dimensions, microstructure, microhardness pitting, and corrosion resistance of 21%-Cr FSS. The depth of the welds dropped initially, and then increased with pulse duration. The grain size in the fusion zone and heat-affected zone (HAZ) and the width of the HAZ all increased with average power, but changed little with pulse duration. The microhardness of welds decreased with the average power. The pitting corrosion resistance of welded joints was weakened with increase in pulse duration. Amuda [5] characterized cryogenically cooled FSS (AISI 430) welds in terms of grain structure and hardness distributions along transverse and thickness directions. Cryogenic cooling reduces the weld dimensions by more than 30% and provides grain refinement of almost 45% compared with conventional welds. Ambade [6] studied the effect of the number of welding passes on microstructure and mechanical properties of 409M FSS. Microstructures in transverse sections were observed with an optical microscope and, with an increasing number of passes, grain growth and the width of HAZ increased. The results of tensile tests revealed that, as number of passes increased, there was a reduction in tensile strength and ductility. Zhou [7] used ER308L austenite filler metal to prepare an AISI430 ferrite stainless steel joint via cold metal transition welding. The microhardness values increased from the weld metal to the coarse grain zone of HAZ, and decreased significantly in the fine grain zone. The microhardness of the base material was lower than that of the weld and HAZ. The welded joints showed good tensile properties and the fracture was located at the base metal for all samples. The fracture surface with uniformly distributed dimples indicated a typical ductile fracture. Khan [8] developed and adopted a combined pre-weld and post-weld treatment to alleviate the problem of microcrack formation. Ramesh Kumar S. [9] used plasma arc welding (PAW) to weld austenitic and FSS. The joining of ferritic and austenitic stainless steels using PAW had no major defects and had around 825 MPa and 72% of tensile strength and elongation, respectively. The microstructure is more columnar in the ferrite regions of butt-welded sections and the mechanical properties of the weld show good strength in the weld region. Ranjbarnodeh [10] studied the microstructure of AISI409 FSS welded with tungsten inert gas (TIG) via electron backscattering diffraction (EBSD), and found that the matrix metal was not completely recrystallized. Lala [11] used continuous laser welding to weld 26Cr-3.5Mo ferrite stainless steel, and obtained a narrower HAZ with no significant grain growth, which improved the tensile strength of the welded joint. Hongxia [12] experimentally concluded that the size of the welding heat input affected the mechanical properties of the welded joint, while the use of pulsed laser welding refined the joint organization and improved the overall performance of the joint. Although laser welding is widely used to improve the mechanical properties of welded joints in FSS, most of the studies have focused on the overall mechanical properties of the welded joint, and few reports are related to the plasticity regulation of the welded joints.

Here, 1.2 mm 441 FSS sheets were used. Conditions of the irradiance (Power/Area) were maintained at a constant, as was the welding speed during laser welding. The plasticity of the welded joint was regulated by changing the diameter of the laser beam spot. Microstructure, tensile properties, and spinning of the welded joint were analyzed. The effects of spot diameter on the plasticity of the welded joint are discussed in terms of elongation after fracture and the reduction of area. The purpose of this study was to reduce cracking of welded joints by regulating plasticity, and to thereby provide a basis for increased commercial applications of FSS.

## 2. Experimental

### 2.1. Test Material

The test material was 441 FSS, with the chemical composition shown in Table 1 and a microstructure shown in Figure 1. The specimen size was 420 mm × 250 mm × 1.2 mm. The steel plate was welded using a HL4006D model Nd: YAG solid-state continuous laser produced by Trumpf company in Ditzingen, Germany, with a rated output power of P = 4.0 kW and a laser beam wavelength λ = 1064 nm, spot diameter D = 0.6 mm, and the butt gap of 0.8 mm. No additional filler material was employed for direct material welding. The defocus was set to +2 mm, meaning the focal point was positioned 2 mm above the material. Oxide and oil contamination of the welding end of the plate and its adjacent area were removed before welding. Metallographic specimens were cross-sectioned by wire cutting, sanded, polished, and then etched in a solution of 2 g of FeCl_3_ in 20 mL of H_2_SO_4_ and 40 mL of C_2_H_6_O. Microstructures of the welded joints were observed with metallographic and scanning electron microscopy.

### 2.2. Welding Parameters

The welding parameters are shown in Table 2. Laser beam defocusing was Δf = +2 mm. Argon was used as the shielding gas at a flow rate of 25 L/min. The same welding speed and laser energy were maintained, and only the diameter of the laser beam spot was varied. The spot size was optimized on the flat plate, and the ferrite pipe fittings were welded with the optimum welding parameters. Spinning performance was also tested.

### 2.3. Tensile Testing

Static tensile testing was carried out according to the GB/T228-2002 [13] metal material tensile and GB/T2651-2008 [14] welded joint tensile test methods, using a DDL series universal testing machine (Sinotest Equipment, SINOTEST, Wuxi, China) with an elongation rate of 0.5 mm/min. It has a maximum testing force capacity of up to 300 kN and offers a speed range from 0.01 mm/min to 1000 mm/min. Additionally, it is equipped with both electrical and hydraulic servo control systems and is accompanied by a digital controller. A tensile specimen is depicted in Figure 2.

### 2.4. Spinning Test

By adjusting the diameter of the laser spot, the processing parameters that can increase the plasticity of welded joints of FSS were examined. The pipe fittings were welded with the same parameters, and were reduced and spun. The macro-morphology and microstructure of the pipe fittings after spinning were imaged, and the plasticity of the welded joints were evaluated as a whole.

Spin testing was carried out using parameters shown in Table 3, and D_R_, r_p_, and β’ are the parameters of wheel diameter, fillet radius, and attack angle, respectively. The pipe wall thickness was t = 2.0 mm, the pipe length was L = 295 mm, and the outer diameter of the welded pipe was D_0_ = 125 mm, as shown in Figure 3. The spinning diagram is shown in Figure 4. A visual sketch of the welded pipe after spinning is shown in Figure 5.

## 3. Test Results and Analysis

### 3.1. Microstructure Analysis of Welded Joints

#### 3.1.1. Microstructure of Weld Seam and the Rule of Grain Growth

Figure 6 shows microstructures resulting from laser beam welding with different spot diameters. The 441 FSS is composed of ferritic grains and does not undergo phase transformation, so the welds in Figure 6a–e had the same microstructures of columnar crystals and central equiaxed grains. The size of the columnar crystals in the weld increased and that of the central equiaxed grains gradually decreased with increasing spot diameter. This technique has characteristics of high welding energy density, high energy concentration, and fast heating and cooling. During welding, the weld has the largest temperature gradient, so the crystallization rate of ferritic grains is very slow and almost without constitutional undercooling, which provides favorable conditions for the growth of columnar crystals [15]. During solidification, the temperature gradient decreased, the crystallization rate of ferritic grains increased, constitutional undercooling occurred, and the crystal morphology gradually transformed into cellular and dendrite grains. Equiaxed crystals formed in the center of the weld when it rapidly solidified. In Figure 6c,d, the columnar crystal growth direction in the weld center is almost perpendicular to the weld; in Figure 6a–c, the growth direction angles of the columnar crystals and center crystals in the weld increased. When the laser beam energy and welding speed remained unchanged, the weld width was more affected than the weld depth by the spot diameter, resulting in a fusion line that ran more parallel to the weld center. Columnar crystals in the weld grew toward the molten pool, which indicated that these crystals grew perpendicular to the fusion line direction. Similarly, with the increase in the spot diameter, the laser energy per unit area decreases, the heat conduction trend is strengthened, and the orientation of the fusion line changes. The columnar crystal in the center of the weld grows perpendicular to the fusion line, and the change of columnar crystal orientation and the distribution of low-melting material in the center of the weld is the direct reason for the decrease in tensile strength and plasticity.

As the level of undercooling increases, Equation (1) shows that the nucleation rate of grains will be much greater than the rate of grain growth. Actual crystallization of a weld metal is closely related to the cooling rate of the weld during welding, which is why the weld grain size was smaller when the laser beam spot diameter was minimized. Such conditions promote generation of equiaxed crystal grains in the center of the weld, which can improve the mechanical properties of the weld area and enhance its plasticity.
*Z* = 0.9 (*N*/*G*)^3/4^(1)
where *Z*, *N*, and *G* are the number of grains per unit volume, nucleation, and growth rates of the grains, respectively.

Transformation of columnar crystals into equiaxed grains can be considered according to the Hunt model, and can occur under the conditions of Equation (2):G > 0.617(N_0_)1/3(1 − (ΔT_N_/ΔT)3) ΔT(2)
where N_0_ and ΔT are determined by the specific material composition, and G and ΔT_N_ are determined by the specific welding technology. ΔT is the undercooling of the molten pool, ΔT_N_ is the undercooling of the equiaxed grain nucleation point, N_0_ is the number of nucleation points, and G is the temperature gradient. According to Equation (2) and the metallographic images (Figure 6), as the spot increased, the weld width increased, the temperature gradient G decreased, and the nucleation undercooling ΔT_N_ of the weld metal in the molten pool decreased, which is less conducive to nucleation. The equiaxed grain zone of the joint therefore decreased with an increase in spot diameter.

#### 3.1.2. Microstructure of HAZ and Rule of Grain Growth

Figure 7a–e shows that the thermally affected composition was a single ferritic grain; no significant precipitation phase was found at grain boundaries or within grains. Although 441 FSS is affected by the thermal cycle during welding, the HAZ did not undergo phase transformation and still comprised lamellar ferritic grains. As the spot diameter increased, the weld width increased and the cooling rate of the molten pool slowed down. Under the effects of the thermal cycle, the grain growth trend was clear, particularly closer to the fusion line. Ferritic grains in the fusion- and heat-affected zones grew by epitaxial crystallization with significant directionality. Ferritic grains far away from the fusion zone did not have significant orientations and exhibited a heterogeneous state because of the difference in thermal conductivity in the grown grains. This epitaxial crystallization method of grain growth was not conducive to the plasticity of welded joints.

The intercept method shown in Figure 8 was used to calculate grain sizes in the HAZ. Ten parallel lines of equal length were drawn on a metallographic image and the average grain size of the HAZ was calculated by determining the number of grains intersecting each parallel line, using the image scale as a reference. The grain size of the HAZ was thus measured and grain sizes of ten lines were averaged to limit the error. The grain sizes of the HAZ in the metallographic image were analyzed with Image-Pro Plus 6.0 software. According to ASTM E112 [16], the average grain size of the HAZ was Grade 5 to 6.

HAZ grains were mainly ferritic. Under the effect of the welding thermal cycle, ferritic grains also began to grow and nucleate, and crystallization occurred in an inhomogeneous nucleation manner in the molten or semi-molten state of the base metal. The molten pool near the base metal has a high degree of undercooling. Hence, according to the theory of minimum energy consumption, nucleation of the base metal is preferentially accomplished in the molten or semi-molten state. Therefore, large numbers of nuclei growing in all directions were generated in the HAZ. The first stage of grain growth was completed when the surrounding grains were close to each other and could no longer grow. In the secondary stage, under the continuous welding thermal cycle, the grains competed with each other, leading to the shrinkage or disappearance of some grains, and leaving other grains to occupy the space. An empirical formula describing ferritic grain growth at constant temperature is given by [17]:(3)$$D^{1.97}-D_{0}^{1.97}=Aexp(-Q_{app}/RT)\Delta t$$
where *D* is the grain size for a holding time at temperature *T*, *D*_0_ is the grain size at *T* = 0, *A* is a material constant, *Q_app_* is the activation energy, *R* is the molar gas constant, and *T* is the temperature. According to Equation (3), ferritic grain growth was limited by the rapid cooling rate of the molten pool because of the decrease in weld width as the spot diameter was reduced.

The welding thermal cycle included two stages where the temperature rose and fell during welding, during which the HAZ of the weld was prone to thermal pinning. There is a large temperature gradient in the HAZ during an actual welding process, i.e., large undercooling causes different parts of the same grain to experience a different temperature. Grains near the fusion line also experience different temperature gradients within the grain; therefore, the grain size near the fusion line was larger than that far from the fusion line in the HAZ. The HAZ grain size increased with a slower cooling rate, because of the wider weld width attributed to the larger spot diameters, as shown in Figure 9.

### 3.2. Tensile Test Results

The static load tensile test on welded joints was performed with different spot diameters. Each group of spot diameters used two samples, and the results were averages of the two samples, as shown in Table 4.

Figure 10 shows the change curves of tensile strength and elongation of welded joints with different spot diameters after fracture. However, the temperature gradient and heat transfer mode of the welding pool did not change significantly, which determined that the orientation of the columnar crystal and the distribution of low-melting point material in the center of the weld were similar, so the strength of the welded joint did not change much when the beam diameter was 0.6–0.73 mm. In Figure 10, the tensile strength of the welded joint generally showed a downward trend with increasing spot diameter, and the maximum tensile strength of the welded joint reached 503.5 MPa when the laser spot diameter was 0.6 mm. When the spot diameter was 0.77 mm, the tensile strength of the welded joint was 342.5 MPa. At the same time, the elongation after fracture of the welded joints also decreased with increased spot diameter. This was shown again in Figure 11. When the spot was 0.6 mm, the elongation rate of the welded joint after fracture approached 14.5%.

Because the welding input was the same, the spot diameter of 0.6 mm enabled the unit area to contain more laser energy, and the narrower weld width increased the heating and cooling speed of the weld. This made the columnar crystal size of the weld smaller and more prone to the phenomenon of component sub-cooling, making the equiaxial dendrite structure easier to generate during weld solidification [18], as shown in Figure 6. With the increase in spot size, the columnar crystal size of the welded joint also increased significantly. Larger grain size means smaller grain boundary areas, which have a lower inhibition effect on dislocations from external forces. Therefore, different spot sizes changed the weld section size, grain sizes, and grain boundary areas, which were the main reasons for differences in tensile strength and elongation after fractures of welded joints. 

With the increase in spot diameter, the fracture surface shrinkage of the welded joint decreased gradually, as shown in Figure 12. When the spot diameter was 0.60 mm, the maximum section shrinkage of the welded joint was 31.5%. When the spot diameter was 0.77 mm, the minimum section shrinkage of the welded joint was 18.5%. The fast welding speed and cooling made it easier for components to be undercooled, resulting in better plasticity in welded joints [19]. With the increase in spot size, the axial grain structure in the weld decreases and the columnar grain size increased, and grain hardness was greater than that at the grain boundary. Plastic deformation often starts at the grain boundary and continues to slip along the grain boundary, so large grain size will inevitably reduce the fracture surface shrinkage of welded joints.

From the above analysis, the welded joints with a spot diameter of 0.60 mm had the best tensile properties. Tensile strength, elongation after fracture, and reduction of area were 503.5 MPa, 14.5%, and 30.5%, respectively. The welded joints with a spot diameter of 0.77 mm had the worst tensile properties. Tensile strength, elongation after fracture, and fracture surface shrinkage were 342.5 MPa, 5.3%, and 18.5%, respectively.

The results of scanning electron microscopy analysis of the tensile fracture of welded joints were shown in Figure 13. In Figure 13a,b, the tensile fracture was accompanied by multiple fracture pits of greater depth. The dense, small equiaxial dimples remain at the top of the overlapping crests. These dimples were small in area but evenly distributed, and mainly formed under normal stress in three spatial directions. Some small dimples were attached to the inside of the large dimples, showing the equiaxial characteristic. The small and large dimples combined freely to form the fracture morphology [20]. With the combination of dimples and the increase in area and depth, the elongation and overall plasticity of A4-1 welded joint specimens after fracture were enhanced significantly. The fracture morphology in Figure 13c–e was more inclined to dissociation characteristics, and the dimples aggregation behavior gradually disappeared, belonging to the brittle fracture mode. The pattern of river and fan as well as accumulation of delamination cracks were clearly observed in the specimen fracture [21]. When the weld samples of A4-4 and A4-5 were subjected to tensile action and overload fracture occurs, the ferritic structure crack mostly at the grain boundary and developed rapidly along the direction of load action, forming a lateral delamination structure. In Figure 13f, the parallel step-like river branching patterns were also captured in the fracture image, which further determined the brittle fracture characteristics of sample A4-5. Dislocation accumulation in the cross section blocked the intersection of the sliding surface, resulting in twin accumulation at the grain boundary to produce a large stress concentration phenomenon. In addition, the increase in the spot diameter made the center of the weld extend the coarse columnar crystal structure, and the residual stress after welding was difficult to release through plastic deformation, resulting in the rapid formation of the initial crack on the surface of the dissociation. The crack propagation speed inside the metal matrix was also accelerated, and finally the tensile strength and elongation decreased significantly.

### 3.3. Spin Test Results

The welded and spun tubular parts are shown in Figure 14. The welds appeared silver or light yellow, and no welding defects, such as pores or cracks on the surface, were observed. The tubular parts had almost no deformations and the welds were uniform and very narrow, which is typical of laser welding. The spun surfaces of these parts appeared very natural. They were smooth and without typical spinning defects, such as peeling, ripples, bulges, or cracks. The laser-welded tubular parts showed good overall spinning performance. Elongation was 44.9% and the plasticity was good.

The tubular part welded with a 0.60 mm spot diameter was subjected to axial tangential stress, radial compressive stress, and tangential circumferential stress during spinning. As shown in Figure 15b, the base material and weld grains formed a flow dynamic under the combined action of these forces. The welded part underwent large plastic deformation during spinning. Figure 16b shows that the equiaxed grains and columnar crystals in the weld center broke under the action of the spinning pressure of the roller, and broken ferritic grains rearranged along the spinning direction. With continuous spinning pressure [22], the ferritic grains that had been broken into fine particles were uniformly distributed within the FSS. The continuous action of the spinning roller gradually caused a force in the direction of the wall thickness of the welded tubular parts to become uniform, and plastic deformation also became uniform. When the plastic deformation reached a specific level, internal dislocations of FSS increased sharply, increasing the dislocation density and forming smaller sub-crystals where ferritic grain recrystallization occurred [23]. Therefore, during spinning, the welded tubular parts generated fine ferritic deformation grains and several recrystallized grains.

## 4. Conclusions

The tensile test results indicate that with an increase in spot diameter, the tensile strength, elongation at fracture, and cross-sectional reduction of the welded joint decrease. When the spot diameter is 0.60 mm, the highest tensile strength of the welded joint is 503.5 MPa, with elongation at fracture and area reduction of 14.5% and 30.5%, respectively. The corresponding values for the welded joint with a spot diameter of 0.77 mm are 342.5 MPa, 5.6%, and 28.5%. Fracture surface analysis reveals that the plasticity of the welded joint prepared with a 0.60 mm spot diameter is significantly higher than that of the welded joint prepared with a 0.77 mm spot diameter.

## Figures and Tables

**Figure 1 micromachines-14-02072-f001:**
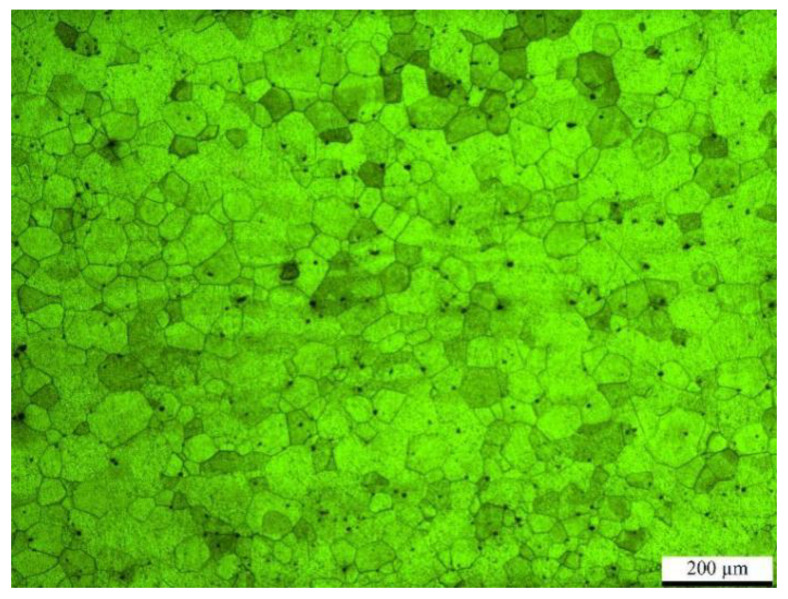
Microstructure of 441 FSS.

**Figure 2 micromachines-14-02072-f002:**
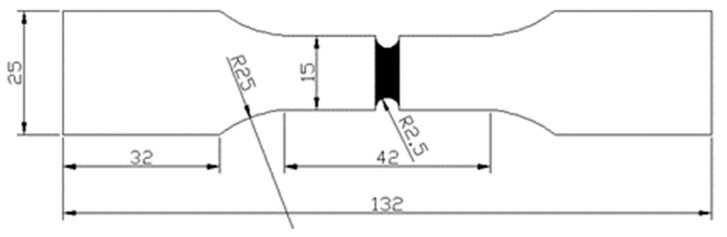
Geometric dimensions of tensile specimen (mm).

**Figure 3 micromachines-14-02072-f003:**
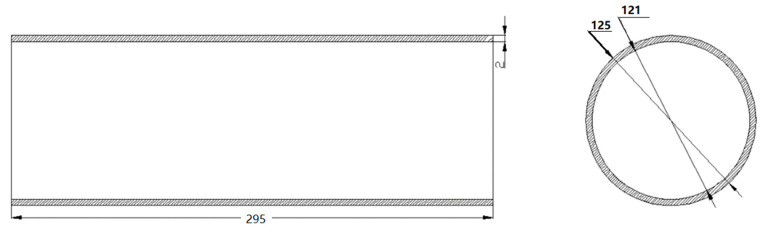
Schematic diagram of spinning tubular part.

**Figure 4 micromachines-14-02072-f004:**
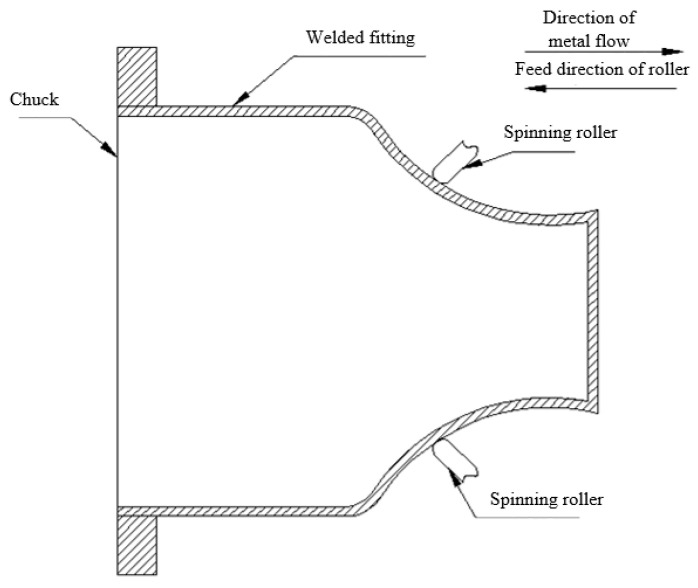
Schematic diagram of spinning.

**Figure 5 micromachines-14-02072-f005:**
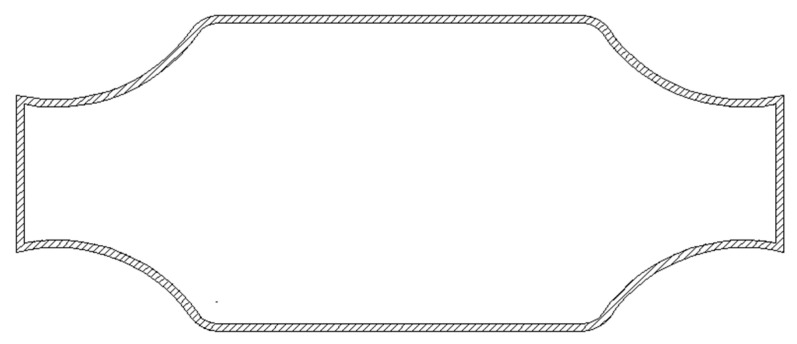
Schematic diagram of welded tubular part after spinning.

**Figure 6 micromachines-14-02072-f006:**
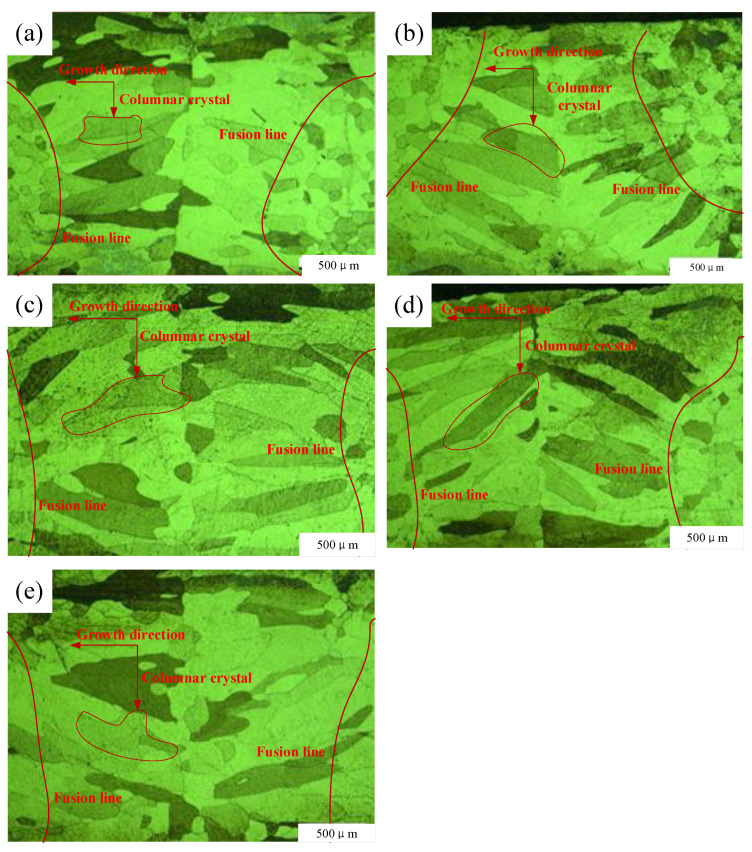
Microstructures of welded joints prepared using spot diameters of (**a**) 0.60 mm, (**b**) 0.64 mm, (**c**) 0.68 mm, (**d**) 0.74 mm, and (**e**) 0.77 mm.

**Figure 7 micromachines-14-02072-f007:**
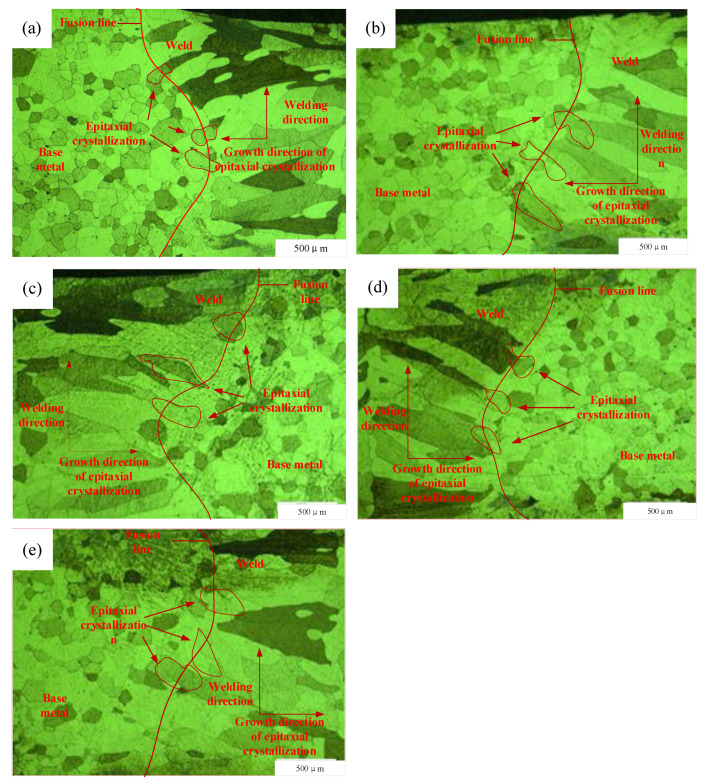
Microstructures of the HAZ of welds prepared at spot diameters of (**a**) 0.60 mm, (**b**) 0.64 mm, (**c**) 0.68 mm, (**d**) 0.74 mm, and (**e**) 0.77 mm.

**Figure 8 micromachines-14-02072-f008:**
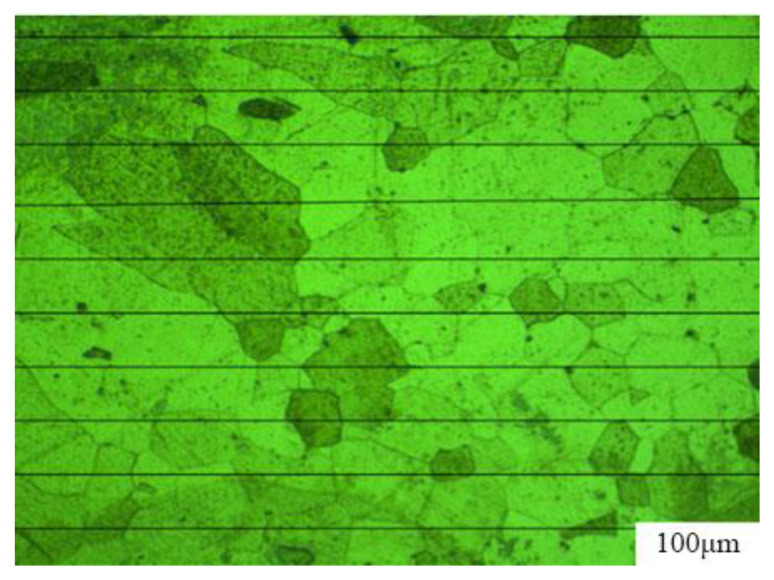
Calculating the grain size using the intercept method.

**Figure 9 micromachines-14-02072-f009:**
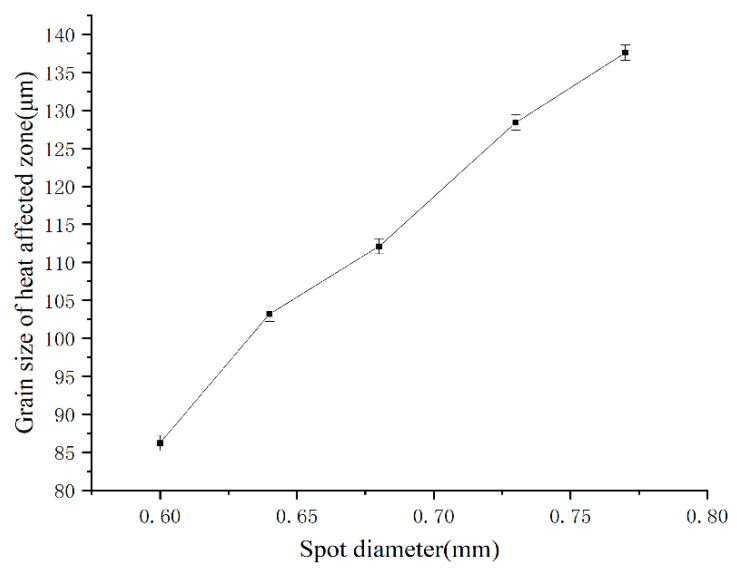
Grain size in the HAZ for different spot diameters.

**Figure 10 micromachines-14-02072-f010:**
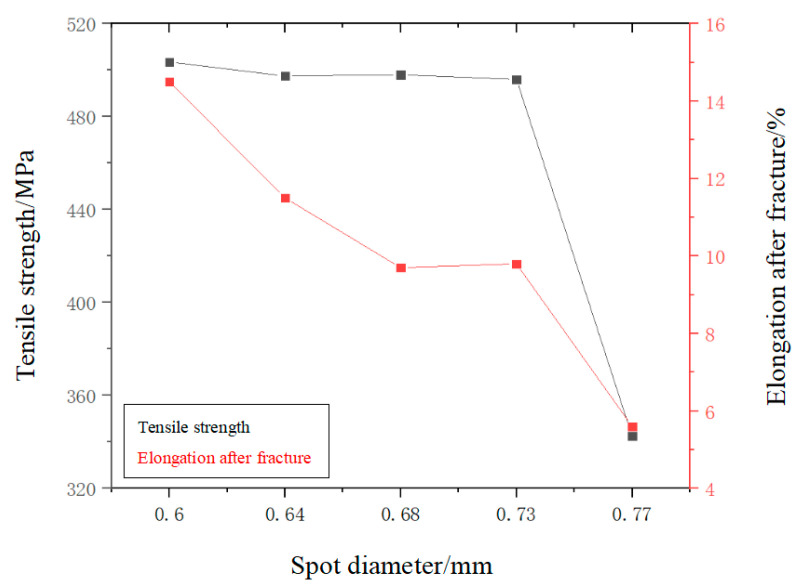
The tensile properties of laser-welded joints with different spot diameters.

**Figure 11 micromachines-14-02072-f011:**
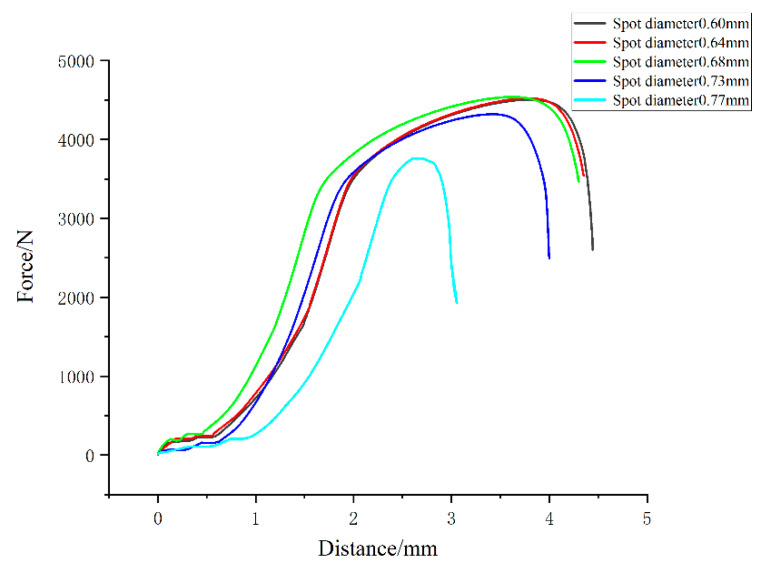
Force as a function of displacement curves for different spot diameters.

**Figure 12 micromachines-14-02072-f012:**
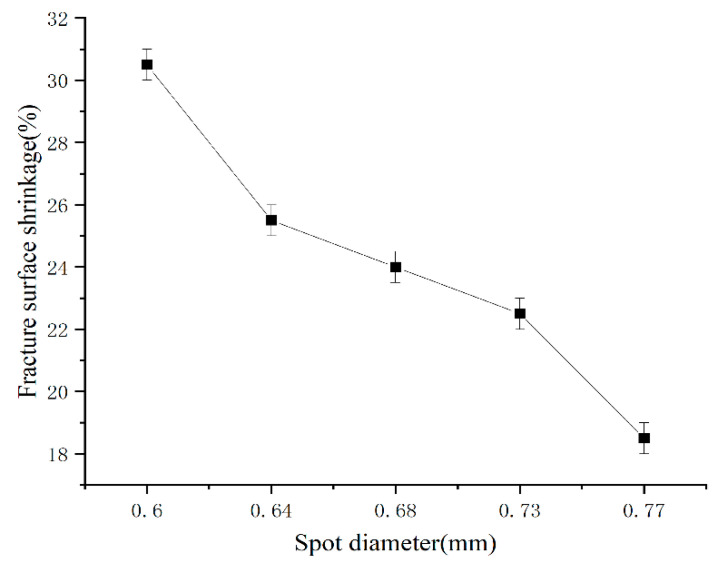
Fracture surface shrinkage of different spot diameters of the welded joint.

**Figure 13 micromachines-14-02072-f013:**
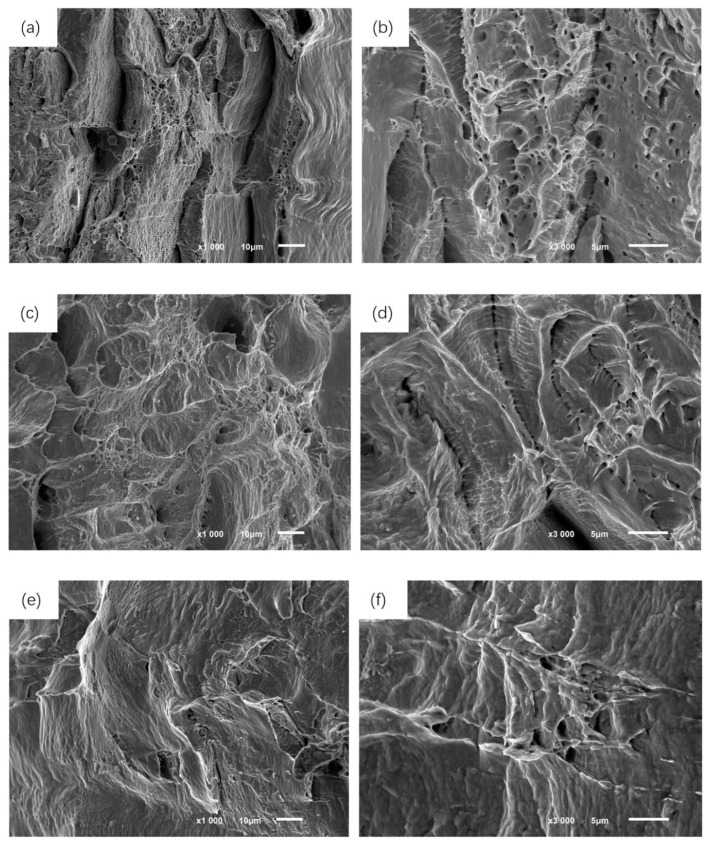
Scanning electron micrograph of weld tensile fracture with typical laser beam diameter. (**a**) 0.60 mm, (**b**) 0.60 mm, (**c**) 0.74 mm, (**d**) 0.74 mm, (**e**) 0.77 mm, (**f**) 0.77 mm.

**Figure 14 micromachines-14-02072-f014:**
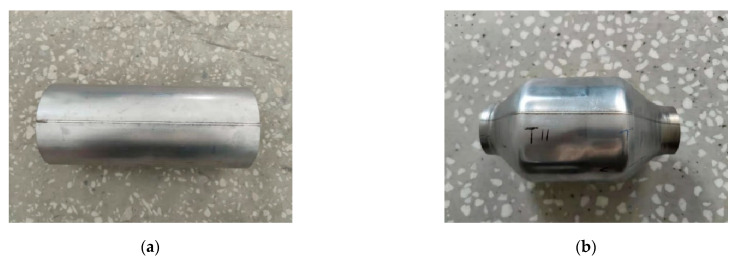
Appearance of laser-welded tubular parts prepared with 0.60 mm spot diameter (**a**) before and (**b**) after spinning.

**Figure 15 micromachines-14-02072-f015:**
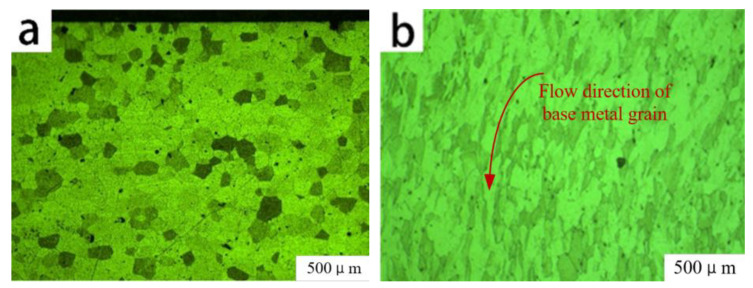
Comparison of base material of 0.6 mm spot laser-welded tubular part (**a**) before and (**b**) after spinning.

**Figure 16 micromachines-14-02072-f016:**
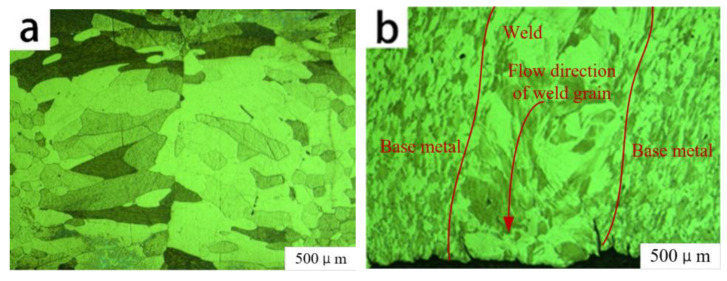
Comparison of weld prepared by 0.60 mm spot laser welding (**a**) before and (**b**) after spinning.

**Table 1 micromachines-14-02072-t001:** Main chemical composition of 441 FSS.

C	Mn	Si	Cr	Ti	Nb	Mo	P
0.03	0.7	0.2–0.8	17.5–18.5	0.1–0.5	0.3 + 3C − 0.9	0.5	0.03

**Table 2 micromachines-14-02072-t002:** Main test parameters.

Serial Number	Laser Area Energy kW/mm^2^	Welding Speed m/min	Spot Diameter mm
A4-1	5.5	1.8	0.60
A4-2	5.5	1.8	0.64
A4-3	5.5	1.8	0.68
A4-4	5.5	1.8	0.73
A4-5	5.5	1.8	0.77

**Table 3 micromachines-14-02072-t003:** Processing parameters of reduction spinning.

Parameter	Welded Pipe D_0_			Spinning Roller D_R_			Feed Ratio	Spindle Speed
	mm t	mm L	mm	mm r_p_	mm β’	(°)	f/(mm/r)	n/(r/min)
numerical	125	2.0	295	200	18	90	1.2	450

**Table 4 micromachines-14-02072-t004:** Tensile test results.

Serial Number	Tensile Strength (MPa)	Yield Strength (MPa)	Elongation after Fracture (%)	Maximum Force (kN)	Fracture Location
base material	435	267	28.9	6.37	base material
A4-1	503.5	420	14.5	4.39	weld
A4-2	497.5	418	11.5	4.47	weld
A4-3	498	407	9.7	4.41	weld
A4-4	496	386.5	9.8	4.46	weld
A4-5	342.5	269.5	5.3	3.08	weld

## Data Availability

The data presented in this study are available on request from the corresponding author.

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
