# Peer review of "Improving Plasticity of Ferritic Stainless Steel Welded Joints via Laser Spot Control"

_micromachines, 2023, doi:10.3390/mi14112072_

Round 1

Reviewer 1 Report

Comments and Suggestions for Authors

In this manuscript, the authors investigated the plasticity of 441 ferritic stainless steel affected by variations in laser spot size. The tensile strength, yield strength, elongation after fraction have been characterized and analyzed based on the change in grain formation in the weld region, base material region, and heat-affected zone. After the spinning process, the welded joints retain good plasticity. The obtained results provide a good data set for study in ferritic stainless steel and match the interests of the broad readership of Micromachines. The manuscript is organized well overall. However, there is some crucial information missing. Before this manuscript becomes acceptable, the authors are required to address all the following comments well.

1. Please provide the laser parameters, including wavelength and pulse duration (if it’s a pulsed laser).

2. In line 84-85, FeCl3, H2SO4, and C2H6O should be FeCl3, H2SO4, and C2H6O.

3. Please explain how you control the laser area energy and beam size.

4. Please explain the definition of DR, rp, β’.

5. The resolution of Fig. 6, 7 should be improved.

6. In line 163-168, N0 should be N0, and ΔTN should be ΔTN.

7. The font format is inconsistent in Table 4.

8. In Table 4, KN should be kN.

9. The unit of distance in Fig. 11 should be mm.

10. In Fig. 13, images a, c, e have a different scale bar or magnification time from images b, d, f. What’s the reason for such a configuration?

11. In line 300, “(a) 0.6” is bolded.

12. Please make correct statements in Conflicts of Interest.

Based on the abovementioned comments, this manuscript is recommended for major revision. A revised manuscript is required.

Reviewer 2 Report

Comments and Suggestions for Authors

The paper " Improving plasticity of ferritic stainless steel welded joints via laser spot control” by Gu et al deals with the influence of laser welding of ferritic stainless steel on its plasticity. The microstructure of welds produced under different defocussing distances was assessed. Moreover, the impact on the mechanical properties and plasticity was determined

The results are in the scope of the Micromachines journal. These can be useful, but I find some points to be addressed:

GENERAL COMMENTS:

This work is interesting and well-written. It deals with a topic relevant. However, after reading the paper I have some general comments:

A)   The main problem is that most of the results are evident. The microstructure is the expected. As the irradiance is lower, the thermal modifications are less aggressive. In view of this, the main novelty is the influence on plasticity. Authors should highlight this aspect.

B)   Conclusions section should be more elaborated. Currently is only a collection of the main results. Conclusions section should answer the main question posted in the introduction (the main aim of the manuscript). In this case, it is the influence on plasticity. Please, rewrite the introduction.

PARTICULAR COMMENTS:

  1. (Line 70) Authors state that “constant laser area energy” was used. In fact, the irradiance (Power/Area) was maintained constant. Please, use the correct terminology.
  2. (Section 2) Please, include the wavelength of the laser source. Was used CW or pulsed laser radiation? I guess that pulsed laser radiation was used as a Nd:YAG laser was used during the experiments. Please, include the pulse frequency and pulse length. Clarify also if any filler was used. Finally, please, include the focal length, clarify that del_ft=+2 mm means that the focal plane is located above the surface of the workpiece.
  3. (Line 100) Which tensile testing machine was used. Include model, company and country of manufacture.
  4. (Subsection 2.4) Why was the spinning test used to measure the plasticity and not others such as Taylor impact test?
  5. (Line 145) It is stated that the growth pattern may affect plasticity of the weld. Please, explain how.
  6. (Lines 158 and 162) Please, define Z in eq. (1) and provide suitable references for both equations. Also, the subindex of some parameters for equations (1)-(2) are missing (Delta_T_N, N_0) in the main text. This also happens for parameters Q_app and D_0 in eq. (3).
  7. (Fig. 9) Please, add error bars to Fig. 9.
  8. (Table 4) Please, replace “KN” with “kN”. Upper-case k means Kelvin, not kilo.
  9. (Fig. 10) Please, explain why the tensile strength under 0.73 mm spot diameter remains almost constant. Can the experimental errors explain this constant trend? Please, add also error bars.
  10. (Fig. 12) Please, add error bars.
Comments on the Quality of English Language

English is fine. 

Reviewer 3 Report

Comments and Suggestions for Authors

In the present research, the authors try to investigate the effect of laser joining spot on the plasticity of ferritic stainless steel welded joints. Though the authors have provided some results, there are still some questions.

1. The authors titled the paper as “Improving plasticity of ferritic stainless steel welded joints via laser spot control”. It seems that the improving in ductility of the ferritic stainless steel weld joint would experience complicated experimental and parameters control. Actually, the content demonstrate the smaller laser spot, the higher ductility for the laser welded ferritic stainless steel. It wonders whether the ductility of the joint would increase further when the laser spot decrease further? Whether does the energy input have no influence on the joint quality? The authors could extend the laser welding spot further to explore its influence.

2. In the experimental, it wonders why the authors have set a indent shape along the welded joint in the tensile specimen?

3. Due to the small laser spot, the cooling rate of the welded region would be high, which might lead to the phase transformation. The authors have not performed the XRD, EBSD and TEM observation in the present research. How could they determine the state of the phase just by the optical microscopy? They are suggested to refer the researches “Improving superficial microstructure and properties of the laser-processed ultrathin kerf in Ti-6Al-4V alloy by water-jet guiding. Journal of Materials Science & Technology 2023,156, 32-53” and “Effect of Synchronized Laser Shock Peening on Decreasing Defects and Improving Microstructures of Ti-6Al-4V Laser Joint. Materials 2023,16 (13), 4570”.

4. In the Fig.9, what is the meaning of grain size? Does it mean the average grain size? If it is such meaning, the error bars should be added.

5. In the content, it wonders whether the authors just test two specimens? If that is, at least one more specimen should be tested for tensile test. Whatever, the error bars should be added for the Fig.10 and Fig.12.

6. In the content, the authors have given the fracture surface of the tensile specimen. Though it could demonstrate some features, however, the effect of welding joint on microcrack propagation could be obtained. The cross-sectional microstructure adjacent to the fracture could demonstrate some feature and information. The authors could refer the recent researches “Tensile, creep behavior and microstructure evolution of an as-cast Ni-based K417G polycrystalline superalloy. Journal of Materials Science & Technology 2018,34 (10), 1805-1816” and “Anomalous yield and intermediate temperature brittleness behaviors of directionally solidified nickel-based superalloy. Transactions of Nonferrous Metals Society of China 2013,24 (3), 673-681”.

7. In the content, there are some spelling errors. The authors are suggested to check the content and revise them.

Round 2

Reviewer 1 Report

Comments and Suggestions for Authors

The authors have answered all comments well. There are some concerns left in the revised manuscript. Please carefully address them.

1. In equation 2, N0 and TN should be N0 and TN.

2. The format of the figure index in Fig.13, 15, and 16 is inconsistent with the rest.

Reviewer 2 Report

Comments and Suggestions for Authors

In the current state, the manuscript entitled "Improving plasticity of ferritic stainless steel welded joints via laser spot control" by Gu et al can be accepted for publication. Authors have addressed all the comments made by the reviewer.

Comments on the Quality of English Language

In the current state, the manuscript entitled "Improving plasticity of ferritic stainless steel welded joints via laser spot control" by Gu requires some minor correction of the English language. Text is easy to follow by some corrections of the English language would improve the quality of the manuscript. 

Reviewer 3 Report

Comments and Suggestions for Authors

The authors have revised the manuscript and answered the question. The paper is improve and could be accepted.